# Structural Transitions and Stability of FAPbI_3_ and MAPbI_3_: The Role of Interstitial Water

**DOI:** 10.3390/nano11061610

**Published:** 2021-06-18

**Authors:** Francesco Cordero, Floriana Craciun, Anna Maria Paoletti, Gloria Zanotti

**Affiliations:** 1Istituto di Struttura della Materia-CNR (ISM-CNR), Area della Ricerca di Roma, Tor Vergata, Via del Fosso del Cavaliere 100, I-00133 Rome, Italy; floriana.craciun@ism.cnr.it; 2Istituto di Struttura della Materia-CNR (ISM-CNR), Area della Ricerca di Roma 1, Via Salaria Km 29.300, Monterotondo Scalo, I-00015 Rome, Italy; annamaria.paoletti@ism.cnr.it (A.M.P.); gloria.zanotti@ism.cnr.it (G.Z.)

**Keywords:** hybrid perovskites, FAPI, MAPI, kinetics of phase transformations, interstitial water, anelastic properties, dielectric properties

## Abstract

We studied the influence of water on the structural stability and transformations of MAPI and FAPI by anelastic and dielectric spectroscopies under various temperature and H${}_{2}$O partial pressure protocols. Before discussing the new results in terms of interstitial water in MAPI and FAPI, the literature is briefly reviewed, in search of other studies and evidences on interstitial water in hybrid halide perovskites. In hydrated MAPI, the elastic anomaly between the cubic α and tetragonal β phases may be depressed by more than 50%, demonstrating that there are H${}_{2}$O molecules dispersed in the perovskite lattice in interstitial form, that hinder the long range tilting of the PbI${}_{6}$ octahedra. Instead, in FAPI, interstitial water accelerates in both senses the reconstructive transformations between 3D α and 1D δ phases, which is useful during the crystallization of the α phase. On the other hand, the interstitial H${}_{2}$O molecules increase the effective size of the MA and FA cations to which are bonded, shifting the thermodynamic equilibrium from the compact perovskite structure to the open δ and hydrated phases of loosely bonded chains of PbI${}_{6}$ octahedra. For this reason, when fabricating devices based on hybrid metal-organic halide perovskites, it is important to reduce the content of interstitial water as much as possible before encapsulation.

## 1. Introduction

Though known since the 1980s [1,2], the hybrid metal-organic halide perovskites like MAPbI${}_{3}$ (MAPI, MA = methylammonium, [CH${}_{3}$NH${}_{3}$]${}^{+}$) have aroused great interest in the scientific community only recently, after finding that they possess the whole set of properties necessary for efficient photovoltaic power conversion [3,4]. In fact, these materials have large absorption coefficients, broad absorption spectra, photocarriers with high mobilities and long lifetimes, so opening perspectives of cheap solar cells with photoactive materials prepared by solution-based chemical methods. In addition, also thanks to the wide tunability of the bandgap with composition, their properties may be exploited in optoelectronic applications, such as LEDs of any color [5] and sensors from infrared [6] to γ-rays [7].

Among the hybrid halide perovskites, FAPbI${}_{3}$ (FAPI, FA = formamidinium, [CH(NH${}_{2}$)${}_{2}$]${}^{+}$) has the second smallest bandgap and highest photovoltaic conversion efficiency under sunlight (FASnI${}_{3}$ has smaller bandgap but also severe stability issues). Unfortunately, the photoactive perovskite phase, often called α-FAPI [8], is particularly prone to the conversion to a photoinactive polymorph δ-FAPI, to degradation and decomposition under the influence of humidity, light and heat, all critical issues for a solar cell. The reported lifetime of α-FAPI is generally of the order of hours [9,10] or days [11,12], and this is also considered an intrinsic problem of the α phase, since the δ phase is calculated as more stable and indeed it is the phase that is generally obtained with chemical methods. Several strategies are attempted in order to improve the stability of α-FAPI [13]; the most explored is the partial substitution of FA with MA and Cs and/or I with Br or Cl, all of which increase the bandgap and are detrimental to the power conversion efficiency. In addition, these mixed halide perovskites are subjected to ionic diffusion and segregation [14], and other degradation paths under illumination, especially for Br/I mixing [15]. Exposure to humidity is a major cause of fast degradation, a fact attributed to the aqueous solubility of the MAI and FAI salts [16,17]. On the other hand, it has been found that α-FAPI films are perfectly stable for at least 90 days if stored in low humidity [18] and the loose powder is stable for months even in air, since water uptake occurs mainly at the grain boundaries, which are absent in the powder [19]. Other strategies for improving the stability are therefore based on additives that should passivate the grain boundaries or various types of encapsulation [13], but the improvements in stability are generally far from the requirements for commercial applications. Yet, it has been recently demonstrated that the addition of 2-(Dimethylamino)ethylmethacrylate to the solvent encapsulates each grain, making the α-FAPI film perfectly stable for at least 100 days in air [20], and considerable stability is obtained also by passivating the iodide vacancies with the addition of formate [21].

Much attention has been devoted to the degradation induced by humidity [22,23,24,25,26,27,28,29], but much less to the catalytic effects of intercalated water molecules on crystallization and polymorphism [19]. Actually, beneficial effects of moisture on the crystal morphology and perfection, resulting in improved photovoltaic efficiency and stability, have been often reported [30,31,32,33,34,35,36,37,38,39], but remain generally unexplained.

Here, we revisit our studies on FAPI and MAPI, based on anelastic, dielectric and X-ray diffraction measurements [19,40,41,42], in order to provide insight in the very important issue of the stability of the 3D perovskite phases of MAPI and FAPI and the multiple roles of water in catalyzing decomposition but also the conversions between polymorphs and crystallization. We also analyze unpublished results that we found at first puzzling, but interpret now as evidence of interstitial water in the perovskite and δ phases. To this aim, the experiments are chosen in order to highlight the role of water in enhancing the kinetics of the δ→α transition in FAPI and hindering the α→β transition in MAPI.

The analysis of the results is accompanied by a review of the literature relevant to the discussed issues. In particular, Section 4.1 deals with the role of the ionic sizes in determining the stability of the perovskite ABX${}_{3}$ structure, showing that an excessive size of A favours the polymorphs where the BX${}_{6}$ octahedra are connected into chains (δ phases), whose separation is free to adjust to the A size. This is the case of FAPI but not of MAPI, with smaller MA cation. Section 4.2 reviews the literature on water and the hybrid perovskite halides, which deals mainly with surface reactions and degradation, more rarely with stoichiometric hydrated phases and never with interstitial water, though some results are suggested to demonstrate its existence. The different effects of hydration on the elastic response of α-MAPI and δ-FAPI are then discussed in Section 4.3 and Section 4.4. When cooling α-MAPI, water hinders the tilting of the PbI${}_{6}$ octahedra into the β phase, considered as evidence of the interstitial form of H${}_{2}$O molecules. When heating δ-FAPI, water promotes the transition to the α phase, again interpreted in terms of interstitial H${}_{2}$O molecules, that catalyze the rearrangement of the Pb-I network from 1D connected to 3D connected PbI${}_{6}$ octahedra. The differences between MAPI and FAPI are next discussed in terms of the different sizes of MA and FA (Section 4.5), considering that the even larger size of the hydrated FA·H${}_{2}$O cation favours the open δ phase. In the last sections, conclusions are drawn on the double role of interstitial water in enhancing the kinetics of recombination of the Pb-I network, with beneficial short term effects during synthesis, and in shifting the energy balance from the compact perovskite phase to the open δ and hydrate phases, with consequent degradation in the long term.

## 2. Materials and Methods

Powders of FAPI and MAPI were prepared and characterised as described in Refs. [40,41]. Briefly, δ-FAPI was obtained by precipitation with toluene from a solution of FAI and PbI${}_{2}$ in γ-Butyrolactone; the powder was then centrifuged and desiccated. MAPI was prepared mixing MAI and PbI${}_{2}$ in glacial acetic acid in an ultrasonic bath at ambient temperature; the powder was decanted, filtered, washed with ethanol and n-hexane and finally dried under vacuum at 60 °C. The powders were checked with X-ray diffraction to consist of pure δ-FAPI and β-MAPI and kept in closed bottles filled with N${}_{2}$.

In order to obtain discs of 10 mm diameter and 0.74 mm thick and bars 40 mm long and 0.45–1.1 mm thick for the dielectric and anelastic experiments, the powders were pressed in air for few minutes at 0.4 GPa (bars) or 0.6 GPa (dics), with pressure applied and released gradually over 2–5 min. While preparing the powder in the die, uptake of water from air occurred, especially in cases when the powder had to be mixed first in an agate mortar because it was a mixture of MAPI and KBr (also hygroscopic), or a mixture of powders from old samples (FAPI #6, see Section 3.1) or a first unsuccessful pressing required rehomogeneisation of the powder (MAPI). The water adsorbed at the powder grains was incorporated in the sample during pressing and, in addition, the pressed samples become particularly hygroscopic at the newly formed grain boundaries [41,43,44].

The complex Young’s modulus $E=E^{\prime }-iE^{\prime \prime }$ was measured by electrostatically exciting the free flexural modes of the bars suspended on thin thermocouple wires [45]. The sample holder was at the end of an insert in a closed quartz tube with a diameter of 90 mm, with thermal screens that create impedance to the gas flux toward the vacuum pump. Therefore, in conditions of dynamic vacuum and high temperature, the vacuum head measured a pressure lower than that felt by the sample, possibly an order of magnitude lower. The resonance frequencies of the samples are $f\propto \sqrt{E}$ and we show the normalized $E\left(T\right)/E\left(T_{0}\right)=f^{2}\left(T\right)/f^{2}\left(T_{0}\right)$, where $T_{0}$ is chosen as the temperature where *E* reaches its maximum value within the α phase. The elastic energy loss, $Q^{-1}$ = $E^{\prime \prime }/E^{\prime }$, was measured from the width of the resonance peak or from the free decay.

The complex dielectric permittivity $\varepsilon =\varepsilon^{\prime }+i\varepsilon^{\prime \prime }$ was measured with a HP 4284A LCR meter in a modified Linkam Examina Probe with a sealed volume of ∼100 cm${}^{3}$. The standard procedure for removing humidity, flushing the probe with a dry gas evaporated from liquid nitrogen for few minutes at 40 °C, was totally ineffective at removing the water adsorbed and intercalated in the sample [41].

From the point of view of the water partial pressure around the sample, the anelastic experiments were conducted in conditions of high vacuum, unless water was intentionally introduced, while in the dielectric experiments only a little fraction of water could be removed by 10 min flushing with N${}_{2}$, even at T> 400 K (see Figure 3), and much water remained in the cell.

## 3. Results

### 3.1. FAPI

Figure 1, adapted from Figure 4 of Reference [19], presents the relative change of the Young’s modulus and the elastic energy loss versus temperature of two bars of FAPI. Sample FAPI #5, pressed from δ-FAPI, had been measured down to 100 K and kept two days in high vacuum, and therefore was dehydrated with respect to the initial state. The curve in Figure 1 was measured during heating at 1.5 K/min in $10^{-6}$ mbar. The mid-point of the step in the Young’s modulus, due to the δ→α transition, is at 410 K in this dehydrated condition. That the sample transformed into the α phase is confirmed by the subsequent cooling, during which the elastic anomalies at the α→β and β→γ transitions [19,42] appeared. FAPI #7 was another freshly pressed δ-FAPI bar measured during heating again at 1.5 K/min in the same high vacuum equipment filled with 7 mbar water. In these conditions, corresponding to 30% relative humidity (RH) at room temperature, the δ→α transition was shifted about 22 K to lower temperature, and was preceded by a more than three-fold increase of $Q^{-1}$, which was already five times higher than in the dehydrated state of curve 7. After completing the transition, high vacuum ($10^{-6}$ mbar) was established and, as usual under vacuum, during the following cooling the perovskite structure was maintained, as demonstrated by the elastic anomalies at the α→β and β→γ transitions.

Figure 2 compares the previous result on FAPI #7 with other two experiments, where the dynamic vacuum was monitored in order to detect the loss of water during heating. The FAPI #6 bar was obtained pressing a mixture of FAPI powders, 60% δ phase and 40% from a sample that had been transformed into α phase and then stored three weeks in a desiccator. This is the initially most hydrated sample, due to the powder mixing in agate mortar in air and use of three-weeks old bar. In this case, the normalization constant $E_{0}$ was not chosen as *E* in the α phase, but in order to approximately maintain the relative magnitude of *E* with respect to the other samples: E≃ 10.2 GPa in the α phase with respect to ≃11.4 GPa of the other samples. We do not plot the absolute values because there is a large error in their determination, especially due to the nonuniform density along the samples [19]. The initial state is stiffer than in the other two cases, in accordance with the presence of α phase, but apparently not all the sample volume transformed into α phase after heating, as demonstrated by the small amplitude of the elastic anomalies at the α→β and β→γ transitions. It is possible that the nominally α-phase powder from the three-week-old sample was partially degraded; no XRD analysis was done in this case.

The interest in the measurement of FAPI #6 is the highly hydrated initial state with respect to the other two measurements. The warming scan was made at 0.5 K/min continuously pumping and monitoring the pressure, in order to follow the desorption of water. The complete set of data versus time (temperature *T*, *T* rate, pressure *p*, $E/E_{0}$ and $Q^{-1}$) is reported in Figure 3. The initial small peak in $p\left(t\right)$ was due to the usual desorption of water from the vacuum insert and quartz tube, but the loss of water from the sample started around 400 K and lasted for ∼1.5 h, during which a temperature of 410 K was maintained. Such a rise of pressure, exceeding $1\times 10^{-5}$ mbar, is well above the usual rise due to the background desorption from the system during heating and is therefore due to the sample. It can only be water and not decomposition of the sample, which occurs above ∼440 K and leaves material condensed on the colder part of the quartz tube. It follows that when the elastic modulus had a first sharp rise around 360 K, the sample was still hydrated. Due to an accident, the anelastic data were not recorded during this short time interval (shown in dotted line), but the occurrence of the structural transition is evident also from the anomaly in the heating rate. In fact, the sample was suspended in vacuum on thin thermocouple wires and therefore with very poor thermal contact with the heater. As a consequence, its temperature profile presents overshoots and oscillations at each change of the temperature rate of the heater. The peak in dT/dt at 360 K, however, occurred at constant heating rate and was due to the latent heat of the structural transformation, during which the elastic modulus increased of 45%. In Section 4.6, we will suggest an explanation for the re-entrant behaviour of *E*, but the main result is that, although the δ→α transformation appears more structured while heating slowly, about 0.5 K/min, in high vacuum, its onset is shifted to lower temperature for higher content of water in the sample.

We tried to reproduce the last experiment in more controlled conditions, pressing the bar FAPI #11 from δ-FAPI powder, whose XRD pattern contained only peaks of δ-FAPI and none of PbI${}_{2}$. The powder was mixed in an agate mortar for about 15 min for simulating the preparation of the mixed FAPI #6 sample and pressed as usual up to 0.5 GPa after a total exposure of 1 hour to 75% RH at 22 °C. The bar became orange, due to the pressure-induced transformation of the δ polymorph (2H in Ramsdell’s notation) into 4H and 6H polymorphs [19]. The bar was then mounted in the high vacuum insert and pumping started after it had been exposed to air for one hour. As for FAPI #6, heating was conducted at 0.5 K/min, but no increase of pressure over a base value of $1\times 10^{-5}$ mbar was detected, indicating that the content of water was much smaller than in the previous case. Accordingly, the $Q^{-1}\left(T\right)$ curve did not exhibit any anomalous increase but only a down step at the *n*H→α transition, which occurred at a temperature much higher than for FAPI #6, and also 4.5 K higher than in FAPI #7 (1.5 K/min in 7 mbar H${}_{2}$O). Evidently, the preparation in air described above did not introduce significant amounts of water.

The fact that the FAPI #7 measurement in 7 mbar H${}_{2}$O presents the δ→α transition at higher temperature than FAPI #6 in vacuum is only in apparent contradiction with the observation that water favours that transformation. In fact, considering that the sample starts loosing water at a fast rate only above 400 K, particularly important is the initial content of water, mainly determined by the initial phases of the experiment in humid air. FAPI #6 was pressed from old powders mixed in air and contained a larger amount of water than in the other cases. The humid condition in which FAPI #7 has been measured was actually very dry, since a constant water pressure of 7 mbar corresponds to 30% RH at room temperature, but <1% above 370 K, where the transition occurred.

Figure 4 presents the dielectric permittivity of a disc of FAPI measured during a series of temperature scans above and below room temperature. The first three runs have been published in Reference [19] and where immediately followed by the acquisition of an XRD diffractogram in air, where the α phase pattern was accompanied by a minority δ phase pattern. The following day, as usual, the α→δ transformation had proceeded further and the $\varepsilon \left(T\right)$ curves labelled 4 were measured. A partial detachment of the electrodes roughly halved the capacity and therefore the permittivity deduced using the same geometric factor. Curve 4 is rescaled for a more clear visualization, but presumably started with ε≃20.

During heating at 5 K/min, the δ→α transformation occurs above 400 K, as before, but it seems incomplete, since the final permittivity remains far below the original value of 48, after 10 min purging with N${}_{2}$ at 440 K followed by aging at 430 K for 90 min in the closed cell. The interesting feature, absent in the previous measurements, is a clear steplike decrease during cooling at 3 K/min, which looks like the reverse α→δ transformation with a thermal hysteresis of 55 K.

### 3.2. MAPI

Figure 5 presents the Young’s modulus and elastic energy loss of MAPI measured during four temperature cycles. Curves 2 and 3 had been reported in Figure 1 of Reference [40], while curve 1 is the initial heating, when the sample was considerably hydrated, as explained in the previous section. The effect of intercalated water on the elastic properties in MAPI seems at first opposite to that in FAPI, since the modulus at room temperature in the hydrated state is stiffer rather than softer than in dehydrated condition. The reason is that at room temperature MAPI is in the tetragonal β phase with tilted PbI${}_{6}$ octahedra, whereas FAPI is in the α cubic phase, but in both cases the cubic phase is made softer by intercalated water. In perovskite MAPI, the main effect of intercalated water is to reduce the amplitude of the steplike softening at the α/β transition, so that the hydrated β phase remains stiffer.

An even larger hindrance of the α→β transition by intercalated water is observed in a sample made of MAPI mixed with 10% KBr (Figure 6). Also KBr is hygroscopic but has no elastic anomalies, so that the amplitude of the step in $E/E_{0}$ is reduced from −50% to −34% in the dehydrated state, but initially is only −16%.

## 4. Discussion

The main results emerging from the experiments presented above can be summarized as follows: (i) the δ→α transition of FAPI is accelerated by intercalated water and therefore occurs at lower temperature during heating; (ii) also the reverse α→δ transformation, which generally occurs over time intervals of hours or days at room temperature, depending on humidity and sample state, may occur abruptly during cooling, if catalyzed by water; (iii) in perovskite MAPI intercalated water does not change the temperature or kinetics of the α/β transition but depresses the amplitude of the associated elastic anomaly, and therefore hinders the transition.

The catalytic effect of water on the δ→α transition of FAPI has already been described [19], and is further confirmed here, where the onset of the transition has been shifted down to 360 K (FAPI #6 in Figure 2). Also the occurrence of the reverse α→δ transition at once during cooling, rather than over hours or days, had been observed in a neutron diffraction study during slow cooling through ∼300 K [46], and possibly as a small negative step sometimes appearing in the cooling $\varepsilon \left(T\right)$ curves [41]. Yet, curve 4 in Figure 4 shows a nearly complete reverse α→δ transition at 355 K during fast cooling at 3 K/min.

Regarding the qualitatively different effect of water in MAPI reported here, we immediately observe that water has a catalytic role in promoting those structural transitions between polymorphs where a reconstruction of the Pb–I framework is required, namely, between α and δ phases in FAPI, but the effect is opposite between the perovskite polymorphs, like α and β phases in MAPI.

Before arguing that the present observations imply that water must be intercalated in interstitial form both in FAPI and MAPI, we recall the main structural differences between the two compounds and then briefly review the literature, in search of other studies and evidences on interstitial water in MAPI and FAPI.

### 4.1. Ionic Sizes and Differences between MAPI and FAPI

Geometrical considerations based on the empirical ionic sizes are rather useful for classifying inorganic perovskites, but their use becomes much less effective with hybrid perovskites, mainly because of the non-spherical shape of the organic cations and for the H bonds that they form with the surrounding halides. Also stereochemically active cations, such as Pb${}^{2+}$, Sn${}^{2+}$ and especially Ge${}^{2+}$ with their lone pairs [47], may render such simple geometric indicators ineffective. The most popular indicator for the stability of the perovskite ABX${}_{3}$ structure is the tolerance factor [48,49]
(1)$$t=\frac{r_{A}+r_{X}}{\sqrt{2}\left(r_{B}+r_{X}\right)},$$
defined in terms of the effective ionic sizes, usually those tabulated by Shannon [50]. With this definition, the radii perfectly fit the cubic perovskite structure when t=1. When t < 1, the BX${}_{6}$ octahedra are too large to fit the A–X bonds and therefore, if they are also more rigid than the A–X bonds, below a critical value ∼0.9 they tilt without shrinking. This generally occurs during cooling, when the weaker and therefore more anharmonic A–X bonds shrink more than the B–X ones, bringing *t* below the critical value [51]. On the opposite side, above a critical value t≳1, the A cation cannot fit in the cavities between the BX${}_{6}$ octahedra and a series of structures with different octahedral connectivity is favoured [8,52,53]. In this context, it is useful to adopt Ramsdell’s notation [54], according to which the cubic perovskite is called 3C, the δ-FAPI structure is 2H, and the phases with intermediate connectivity are 4H and 6H [55,56,57].

In view of the complications connected with non spherical A cations and H bonds, revisions of the effective ionic radii may be effected, so that *t* of FAPI may be calculated as 0.987 [58] or 1.03 [47] and that of MAPI as 0.912 [58]. Alternatively, new parameters may be devised for predicting the stability of the perovskite structure [59]. In any case, the difference between *t* of MAPI and FAPI is sufficiently large that the main differences in their structural transitions and hence stability and behaviour in the presence of water may be rationalized in terms of the different sizes of MA and FA [19,42].

The MA cations are clearly too small with respect to the cavities between the relatively rigid PbI${}_{6}$ octahedra, which therefore tilt, giving rise to the tetragonal β phase already above room temperature. Instead, the larger FA puts FAPI in a borderline situation where it exhibits the features characteristic of both t<1 and t>1 [19]. In fact, the highly anharmonic shrinking of the weak FA-I bonds determines octahedral tilting on cooling just below room temperature (small *t*). On the other hand, at room temperature a number of other polymorphs characteristic of compounds with too large A cation (large *t*) are slightly more or slightly less stable than the perovskite structure: the 2H (δ), 4H and 6H phases with progressive change of the octahedral connectivity from 1D face-sharing chains to 3D corner-sharing [55,56,57]. The evidence of this is the fact that moderate pressure (0.4 GPa) induces the transformation of the yellow 2H phase into orange 4H and red 6H structures, stable, or at least metastable over a long time span, at room temperature [19]. Another type of reversible transformations among layered perovskite structures, again with colours ranging from yellow to red, has been found in layered Ruddlesden-Popper FA${}_{n+1}$Pb${}_{n}$I${}_{3n+1}$ films exposed to humidity [60]. In that case the change of stoichiometry *n* is proposed to occur through the diffusion of FAI among grains. No appearance of the *n*H perovskite structures was found in that case [60], and we assume that, symmetrically, no layered perosvkites appear as intermediate structures of cubic FAPI.

In the case of MAPI, the smaller size of MA does not favour the 1D- or 2D-connected PbI${}_{6}$ octahedra over 3D perovskite. Even if new structures of MAPI have been theoretically considered, with calculated energies also lower than that of perovskite [61], no stable MAPI polymorphs of that type has been observed up to now. On the other hand, this type of polymorphs is observed in hydrated or otherwise solvated forms, as MAPI·H${}_{2}$O or MAPI·DMF (DMF = dimethylformamide) [22]. In understanding the role of interstitial H${}_{2}$O in MAPI, we find useful the point of view that the molecules of solvent, once H-bonded with MA, form cations larger than MA. These non-perovskite structures of 1D and 2D connected octahedra, also typical of the PbI${}_{2}$ polymorphs [62], play a fundamental role in the formation and degradation of MAPI and FAPI [22,63,64] in the presence of H${}_{2}$O or other solvents.

### 4.2. Hydrated Phases and Interstitial H_2_O

In spite of the importance of humidity on the stability of these hybrid perovskites, there is only scant interest in the literature on interstitial H${}_{2}$O molecules. It is usually assumed that the reactions induced by humidity occur within a condensed liquid phase [24,31,37,65], and also the improved morphology induced by humidity is attributed to dissolution and recrystallization of the grains [31,36,66]. Accordingly, most of the experimental and theoretical studies are focused on the reaction of water at the grain surface [25,67,68,69] (for a more extended bibliography see Reference [44]).

As for the catalytic mechanisms of H${}_{2}$O molecules, we mention the transformation of α-MAPI into PbI${}_{2}$ with loss of MA [25]. The H${}_{2}$O molecules form H bonds with the N-H ends of MA but also with I and the interposition of H${}_{2}$O in the N-H…I H-bond lowers the energy barrier for the reaction ending with PbI${}_{2}$ [25]. On the other hand, it is conceivable that the same mechanism may promote other recombinations, in particular the rearrangements of the PbI${}_{6}$ from edge- or face-sharing of the yellow phases to corner-sharing of the black perovskite phase [19], which is what we wish to demonstrate.

Coming back to the literature, it has been recognized that the degradation through exposure to humidity of MAPI passes through an intermediate reversible step, identified as a hydrated phase with isolated octahedra [70], The monohydrate MAPI·H${}_{2}$O phase is formed of edge-sharing double chains of PbI${}_{6}$ octahedra with intercalated H-bonded MA·H${}_{2}$O molecules [22,71]. It is metastable in air, and looses water transforming into α-MAPI, demonstrating that bonding of H${}_{2}$O to MA and PbI${}_{6}$ octahedra does not necessarily lead to decomposition into PbI${}_{2}$ and loss of MA [23]. It has been further demonstrated that MAPI films exposed to 80% RH transform into the monohydrated phase within the first hour and only after two hours does the dihydrate (MA)${}_{4}$PbI${}_{6}\cdot 2$H${}_{2}$O appear, where the octahedra are isolated from each other [23,72]. Only the second step involves the formation of stable PbI${}_{2}$ and therefore initiates the irreversible decomposition. Other hydrated structures of perovskite related halides have been identified [27] but, to our knowledge, none with the FA cation.

Diffusion of H${}_{2}$O molecules as interstitial species is generally not considered; it is rather assumed that their fast in-diffusion occurs along grain boundaries, or that H${}_{2}$O substitutes the MA cation [38]. On the other hand, there are practically no experiments devoted to studying the possible penetration of H${}_{2}$O in interstitial form in the hybrid perovskite halides. As experimental evidence that the H${}_{2}$O molecules can intercalate in the perovskite lattice we mention the observation by XRD of a lattice expansion in FASnI${}_{3}$ exposed to 60% RH, which is absent after exposure to 10% RH [26]. The penetration of water is supposed to depend on the low crystallinity of the film, but a lattice expansion revealed by XRD rather than a simple change of macroscopic volume, indicates that the H${}_{2}$O molecules are in interstitial form. This observation is at variance with a lattice contraction observed under humid conditions in MAPbCl${}_{3}$ [38], where it is proposed that H${}_{2}$O enters as substitutional defect of MA [38]. On the other hand, MAPbI${}_{3}$ has a larger cell than MAPbCl${}_{3}$, due to the larger iodide, and may host interstitial H${}_{2}$O: an indication in this sense is the reversible photoluminescence blue shift of MAPI under increased partial pressure of water, that has been associated to an increases of the Pb-I bond length due to intercalated water [73].

From the theoretical side, there are various DFT calculations showing that H${}_{2}$O can exothermically enter the perovskite lattice of MAPbX${}_{3}$ and remain in a stable interstitial form [44,67,74,75,76]. Monohydrate MAPI consisting of the cubic perovskite with each cavity between the octahedra containing MA and H${}_{2}$O, as modelled in Reference [67], is not observed, but this does not exclude that a lower concentration of interstitial water in the 3D perovskite is possible. Its activation energy for diffusion has also been calculated as 0.28 eV in MAPI [74], which corresponds to $10^{10}$ s${}^{-1}$ jumps at room temperature, assuming the usual extrapolation to infinite temperature of $10^{14}$ s${}^{-1}$ for point defects. We are not aware of experimental verifications of the diffusion coefficient of interstitial water in MAPI or FAPI, but a mobility as high as this would justify that even small concentrations of water readily catalyze the structural transitions over the whole grain.

### 4.3. The Reduction of the Elastic Softening at the α/β-MAPI Transition and Interstitial H_2_O

The important observation in MAPI is that intercalated water, in addition to making the cubic phase softer as in FAPI, strongly decreases the amplitude of the softening of the tetragonal phase with respect to the cubic one below $T_{\alpha \beta }$ (Figure 5 and Figure 6). The observed reduction of the amplitude of the elastic anomaly by over 50% can only be explained by assuming that the H${}_{2}$O molecules are uniformly distributed over the bulk in interstitial form and hinder the tilt transition of the octahedra. Hydrated phases can be excluded as source of the phenomenon. In fact, if water were confined to presumably softer hydrate phases at the grain surfaces, the effect would be a further softening below $T_{\alpha \beta }$; on the other hand, if these surface phases were so much stiffer to increase the modulus below $T_{\alpha \beta }$ as observed in the initial state, they would stiffen also the cubic phase, which is certainly not the case. Nor can the reduction of the amplitude of the elastic softening be attributed to a reduction of the volume fraction in the β phase, because it would imply that the α/β transition is prevented in half or more of the sample volume, certainly not a surface effect.

It can be concluded that the marked reduction of the softening below $T_{\alpha \beta }$ implies that water is uniformly dispersed in interstitial form and hinders the long range tilt pattern of the octahedra, mainly responsible for the transition.

### 4.4. The Kinetics of the α/δ-FAPI Transition and Interstitial H_2_O

As noted in Results, the observations on the effect of water in FAPI regard the reconstructive transformations among the non-perovskite and perovskite phases, and are of a different type with respect to those in MAPI on the transitions between perovskite phases. The transformation between the FAPI polymorphs involve reconstructions of the Pb–I network and put in evidence the catalytic action of interstitial water on such processes. This is observed both for the δ→α and the reverse transitions. The first case is exemplified by Figure 1 and Figure 4, where the onset of the δ→α transition, accompanied by large stiffening, is shifted to lower temperature at higher hydration level. The case of the re-entrant transition in FAPI #6 of Figure 2 will be discussed in Section 4.6, but the onset of the transition satisfies the rule. Notice that the dielectric permittivity curves were measured under more hydrated conditions than the in the anelastic measurements, because the water initially contained in the sample was trapped in the small volume of the measuring cell, but the temperatures of the δ→α transitions are comparable to those in the anelastic experiments. This is due to the fact that the heating rates in the dielectric experiments were higher (5 K/min rather than 1.5 and 0.5 K/min) and it has already been noted that, due to the slow kinetics of the reconstructive transition [46], the thermal hysteresis between heating and cooling is widened by faster temperature rates [41].

The catalytic action of unwanted water from humid air on the δ→α transition in FAPI strongly suggests that water is dispersed in interstitial form in δ-FAPI and is highly mobile. This is not difficult to accept, since the rigid chains of PbI${}_{6}$ octahedra of the 2H phase pose a strong limit to the expansion of the *c* lattice parameter, but the distance between the chains in the ab plane is free to expand in order to host any type of molecule, just like the hydrated phases of MAPI discussed above. Doubts arise whether these H${}_{2}$O molecules remain in interstitial form also in the perovskite α phase or are expelled to the grain boundaries during the structural transformation. In fact, as explained in Section 4.1, the large size of FA poses FAPI at the limit of stability of the perovskite structure and interstitial water would cause a further increase of the average effective cation size. Yet, the lattice of these metal-organic hybrid compounds is soft and compliant so that, in the absence of contrary experimental or computational evidence, we may suppose that H${}_{2}$O can be hosted in interstitial form also in α-FAPI, up to a certain concentration.

### 4.5. Interstitial Water, Tolerance Factor and Relative Stability of the α and δ Phases

The presence of H${}_{2}$O in interstitial form in both the δ and α phases of FAPI would better explain the enormous asymmetry between heating and cooling generally observed in the kinetics of the α/δ transition, and particularly the range of characteristic times for the α→δ transition at room temperature: from minutes to years, considering that in anhydrous or even low humidity conditions the α phase may be stable for several months [18,20,41]. Moreover, the temperature induced α→δ transformation is seldom observed: a neutron diffraction experiment as a function of temperature at slow rate in a closed volume suggested that $T_{\alpha \to \delta }≃295$ K, but the permittivity curve no. 4 in Figure 4 during cooling at a fast rate of 3 K/min indicates $T_{\alpha \to \delta }$ = 350 K and steps in ε during cooling through the same temperature, though with smaller amplitude, have been already observed (Figure 4 of Reference [41]). The occurrence of the α→δ transition at higher temperature or at a faster rate at room temperature correlate with the expected level of hydration of the sample, even though not quantitatively assessed, but cannot be reconciled with the stability of α-FAPI for months even in air [41] or at low humidity [18], under the hypothesis that water is only adsorbed at the grain surface.

The explanation we propose is that water may also be present interstitially in α-FAPI and, above a critical concentration, makes the perovskite structure unstable in favour of δ-FAPI, with its open structure of 1D connected octahedra. We cannot exclude that the α→δ transition is stimulated by clustering of H${}_{2}$O molecules in neighbouring cells rather than by an increase of the average lattice parameter and tolerance factor above a threshold.

A similar mechanism may be operative in MAPI, namely interstitial water increases the effective MA cation size shifting the thermodynamic equilibrium from the compact perovskite to open structures, where the MA combined with H${}_{2}$O may be better accommodated. The difference with FAPI is that no stable δ-MAPI structure has ever been observed in anhydrous conditions, but, rather, hydrated MAPI phases exist [22,23,70,71,72]. On the other hand, no stable stoichiometrically hydrated form of FAPI is known, but δ-FAPI may certainly accommodate easily interstitial H${}_{2}$O and therefore be considered as a non-stoichiometric hydrate phase.

A calorimetric study of the influence of humidity on the α→δ transition has been conducted in CsPbI${}_{3}$ [77]. It was found that the transformation enthalpy does not depend on the cooling rate or humidity level, so that it is concluded that water catalyzes the transformation but does not change the equilibrium free energies of the two phases. This contradicts the concept of interstitial water that shifts the relative equilibria of the α and δ phases, but it is possible that the cell of CsPbI${}_{3}$, much smaller than those of MAPI and FAPI, cannot accommodate interstitial H${}_{2}$O, because it would cause excessive local strain. Indeed, the instability of perovskite CsPbI${}_{3}$ is due to the opposite reason than in FAPI, namely to excessively small tolerance factor, and the δ phase is different from the polymorphs of FAPI. It would be interesting to repeat that experiment on MAPI and FAPI.

In the context of the relative stability of the α and δ phases depending on the water content, a remark is necessary on the frequent use of the concept of lattice strain relaxation for stabilizing the perovskite phase through mixing different A cations and X anions in order to optimize the tolerance factor [13,78,79]. In pure FAPI or MAPI there is no lattice strain, especially in the perfectly cubic α-FAPI; it is the difference in stress between the A–X and Pb-X bonds that tends to be minimized, for example, through octahedral tilting. In that case, the more rigid B–X bonds change very little their length and the pressure to which they are subjected is released thanks to the enhanced deformation, and therefore stress, of the softer A–X bonds, with a negative overall balance. Alloying with cations of different sizes may optimize *t* and hence minimize stress, but the lattice strain is necessarily enhanced, unless cation or anion ordering in superstructures occurs. Furthermore, the formation of a vacancy in a single cation perovskite [79] certainly does not reduce the lattice strain but creates it; yet, it may decrease the overall stress and therefore energy.

### 4.6. Re-Entrant Behaviour of the δ → α Transition in FAPI and Anelastic Relaxation from the Action of H_2_O

The interpretation of the re-entrant behaviour of *E* in the δ→α transition in Figure 3 is not straightforward, but it should be considered that it is not a simple phase transition describable with an order parameter, like the α/β and β/γ transitions due to octahedral tilting and FA orientational ordering. In fact, the δ/α transition is reconstructive and has been proposed to occur through several intermediate metastable structures [46]. In addition, the stable or very long lived metastable structures with octahedral connectivity intermediate between the 1D face-sharing chains of the δ phase (2H) and the 3D corner-sharing α cubic perovskite (3C) have been identified: the orange 4H and red 6H phases of FAPI [19,55,56]. The slow heating rate and low temperature at which the catalytic action of interstitial water promotes these transformations prolong the process over a wide temperature span, over which the energetic balance between the different phases may change. In addition, during the process water is lost, with the consequence of loosing its catalytic action and slowing the process before it is completed.

The role of water in this prolonged out-of-equilibrium phase is demonstrated by the fact that the enhancement of the anelastic losses coincides with the loss of water from the sample; indeed, the $Q^{-1}\left(T\right)$ and $p\left(T\right)$ curves are very similar to each other and there is an anticorrelation with the $E\left(T\right)$ curve. The H${}_{2}$O out-diffusion and the continuous structural changes occurring in that temperature range cause several anelastic relaxation processes, each of the Debye type $\delta E/E_{0}=\Delta /\left(1+\iota \omega \tau \right)$, where τ is the elementary relaxation time and ω the angular vibration frequency [80,81]. Each elementary relaxation process causes a peak of intensity Δ/2 in the $Q^{-1}$ at the temperature $T_{0}$ at which $\omega \tau \left(T_{0}\right)=1$ is satisfied, and a softening −Δ in the modulus at the temperatures $T>T_{0}$ for which ωτ≤1 [80,81]. The re-entrant relative softening in Figure 3 is nearly −0.4, while the maximum value of the losses is only $Q^{-1}∼0.07$, three times smaller than expected from the softening, but this is normal for broad distributions of relaxation times. In fact, at each temperature only the processes satisfying ωτ∼1 contribute to $Q^{-1}$, while all those with ωτ≤1 contribute to the softening, so that the re-entrant softening of FAPI #6 in Figure 2 and Figure 3 is compatible with a dynamic effect rather than intrinsically softer intermediate state of the material.

A rise of dissipation similar to that found in highly hydrated FAPI #6 occurred also in FAPI #7 during heating in 7 mbar H${}_{2}$O (Figure 2), further corroborating the relationship with interstitial H${}_{2}$O: such excess dissipation is observed only in samples exposed to water, that therefore have a lower δ→α transition temperature.

### 4.7. Reproducibility of the Experiments with Hydrated Samples

A comment on the reproducibility of the experiments with hydrated samples is due. In fact, as apparent from the failure of repeating with sample FAPI #11 the experiment done on FAPI #6, we are not able to introduce controlled amounts of water in our samples, especially during the process of pressing a nominally dehydrated powder in air. We qualitatively note that shortly after pressing a black powder of MAPI or FAPI, yellow or red hues may appear, and we observe strong differences in the kinetics and completeness of the δ→α structural transitions between dehydrated and hydrated samples. The difficulty in controlling the level of hydration arises from the concomitant degradation and decomposition and from the determinant influence of many variables that we do not control, including the density and type of defects at the grain surface and grain boundaries. On the other hand, the behaviours of different hydrated samples are quite consistent with each other, so that qualitative but solid conclusions can be drawn. The differences observed from experiment to experiment are attributable to differences in the level of hydration but, once dehydrated, the materials exhibit reproducible behaviour, and may remain stable for months [41].

### 4.8. Beneficial Role of Water during Preparation

There are several studies that report beneficial effects of moisture on the crystal morphology and perfection, which finally result in improvements of the photovoltaic efficiency and stability of the devices [30,31,32,33,34,35,36,37,38,39,66,82,83]. These effects are generally attributed to liquid phases that promote the dissolution and recrystallization of the grains [31,36,66], but in view of the conclusions reached here on interstitial water, we suggest that also the latter plays a role. Unless steric effects make the free energy of the H${}_{2}$O molecule as interstitial defect too high, possibly in compounds with very small cell size, it is difficult to imagine that, during crystallization from a solution with a more or less intentional content of water, no H${}_{2}$O molecules remain trapped in solid solution in the perovskite and especially in the open non-perovskite structures. An indication in this sense seems the recent observation that a simple double-step preparation of FAPI, where the δ phase is heated in air to form α phase, produces films with much improved microstructure and stability [84]. This may be explained with a particularly efficient recrystallization process during the δ→α transition catalyzed by intercalated water. Another advantage of catalyzing the δ→α transition in preparations that start with δ-FAPI is the possibility of limiting the exposure to high temperature, since FAPI starts decomposing above 430 K [85] and FA above 370 K [13,86].

A positive role of humidity has also been found in the otherwise dry mechanosynthesis of MAPI, where it catalyzes the reaction between MAI and PbI${}_{2}$ [87]. It seems therefore promising to study crystallization processes in presence of controlled amounts of water or humidity, especially in combination with in situ monitoring of the crystallization process, as with GIWAX [39,56,57] or optical reflectance [88].

### 4.9. Removal of Water

Water adsorbed and intercalated in perovskite MAPI and FAPI certainly is detrimental, even in small amounts, to the long term stability, especially under exposure to light [89]. Whether intentionally introduced for improving the quality of the film, or present in unknown amounts in the starting solvents or environment, it must be removed before sealing the device. The complete removal of intercalated water requires high vacuum conditions and high temperature, not only for accelerating its out-diffusion, but also to lower the concentration of interstitial H${}_{2}$O in solid solution, which is in equilibrium with the surrounding atmosphere. The phase diagrams of water intercalated in MAPI or FAPI are not known, but it is likely that the equilibrium concentration of interstitial H${}_{2}$O with a given partial pressure or humidity is a decreasing function of temperature. We are not aware of any systematic study on the removal of water in hybrid halide perovskites, and found only sparse information. For example, a 24 h long exposure in unspecified vacuum is not sufficient to completely remove water from MAPI films prepared in air [90]. The test in Figure 3 indicates that to extract most of the water from a sample 1 mm thick like FAPI #6, it is necessary to treat it in high vacuum for ∼1.5 h at 410 K. For a thin film the out-diffusion of water would certainly be faster.

## 5. Conclusions

Water is known to be the major cause of degradation of the metal-organic halide perovskites MAPI and FAPI through chemical reactions at the surface with an adsorbed liquid phase and is known to form stoichiometric hydrates with MAPI. On the other hand, there is ample literature on the beneficial effects of humidity during the preparation of films in terms of improved morphology and crystallinity, with important enhancements of the photovoltaic efficiency and sometimes material stability. In order to provide insight in these phenomena, we studied compacted powders of perovskite β-MAPI and δ-FAPI where the water adsorbed from humidity is incorporated within the bulk during pressing. The various structural transformations of these samples were then studied by anelastic and dielectric experiments under various temperature and vacuum protocols. It is concluded that water can diffuse in the perovskite and other polymorphs not only along grain boundaries but also in interstitial form. The role of interstitial H${}_{2}$O molecules is twofold: (i) they catalyze the structural transformations between the various stable and metastable polymorphs of FAPI with different connectivity of the Pb-I framework by forming intermediate configurations through H bonds; (ii) they form O…H–N bonds with the MA/FA cation, forming a hydrated cation with increased effective size, that shifts the equilibrium from the closed perovskite structure toward the δ and hydrated phases with loosely bonded chains of face- and edge-sharing BX${}_{6}$ octahedra, where larger cations can be easily accommodated.

The catalytic action on the reconstructive transformations is beneficial during the preparation of the perovskite phase, but on the long term interstitial water favours the formation of the more open structures and must therefore be thoroughly extracted before encapsulating the device.

## Figures and Tables

**Figure 1 nanomaterials-11-01610-f001:**
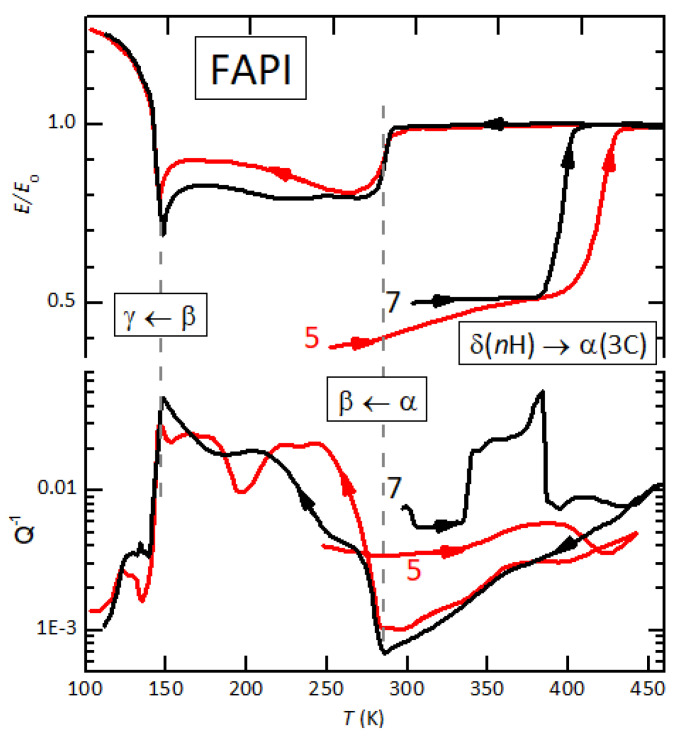
Relative change of the Young’s modulus and elastic energy loss vs *T* of FAPI #5 (dehydrated in high vacuum) and FAPI #7 (heating in 7 mbar H${}_{2}$O). Heating rate: ∼1.5 K/min. Adapted with permission from ref. [19]. Copyright (2020) American Chemical Society.

**Figure 2 nanomaterials-11-01610-f002:**
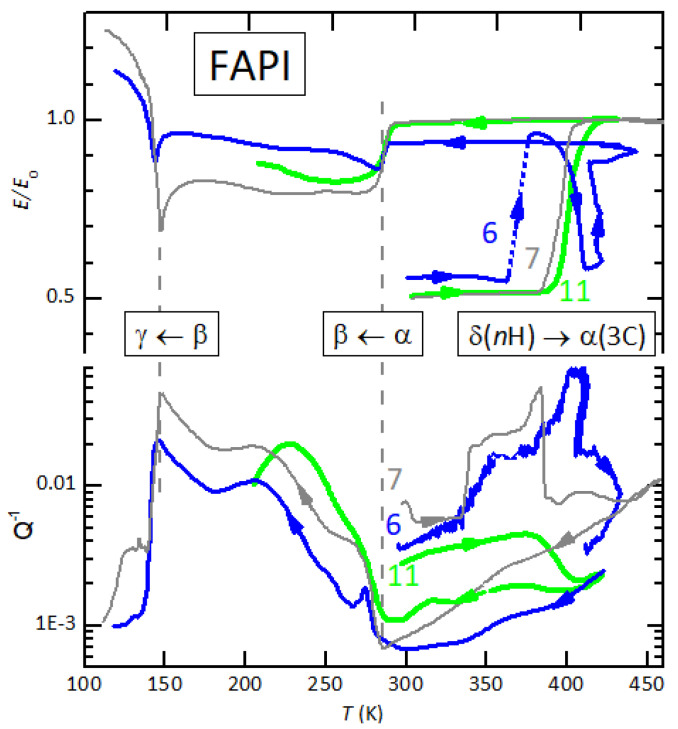
Relative change of the Young’s modulus and elastic energy loss vs. *T* of FAPI #6 (old hydrated powders) and FAPI #11; the heating rate was 0.5 K/min in high vacuum. Furthermore, reported for reference is the measurement of FAPI #7 of Figure 1 (+1.5 K/min in 7 mbar H${}_{2}$O).

**Figure 3 nanomaterials-11-01610-f003:**
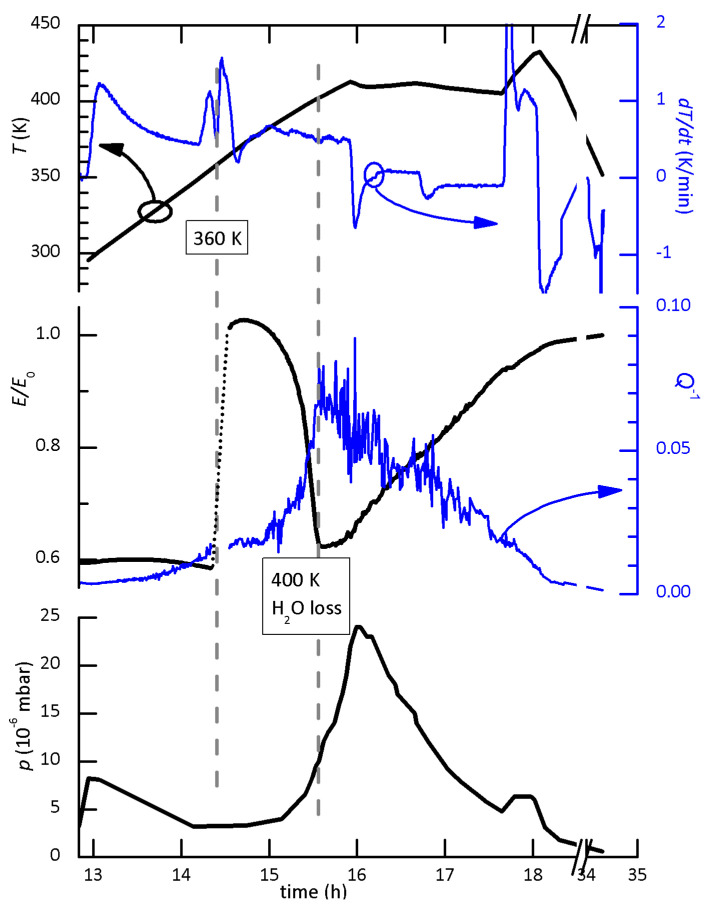
Time dependence of the Young’s modulus $E/E_{0}$ and $Q^{-1}$ of FAPI #6, together with its temperature *T*, dT/dt and pressure under continuous pumping. The pressure around the sample might be an order of magnitude higher than that measured in the vacuum system.

**Figure 4 nanomaterials-11-01610-f004:**
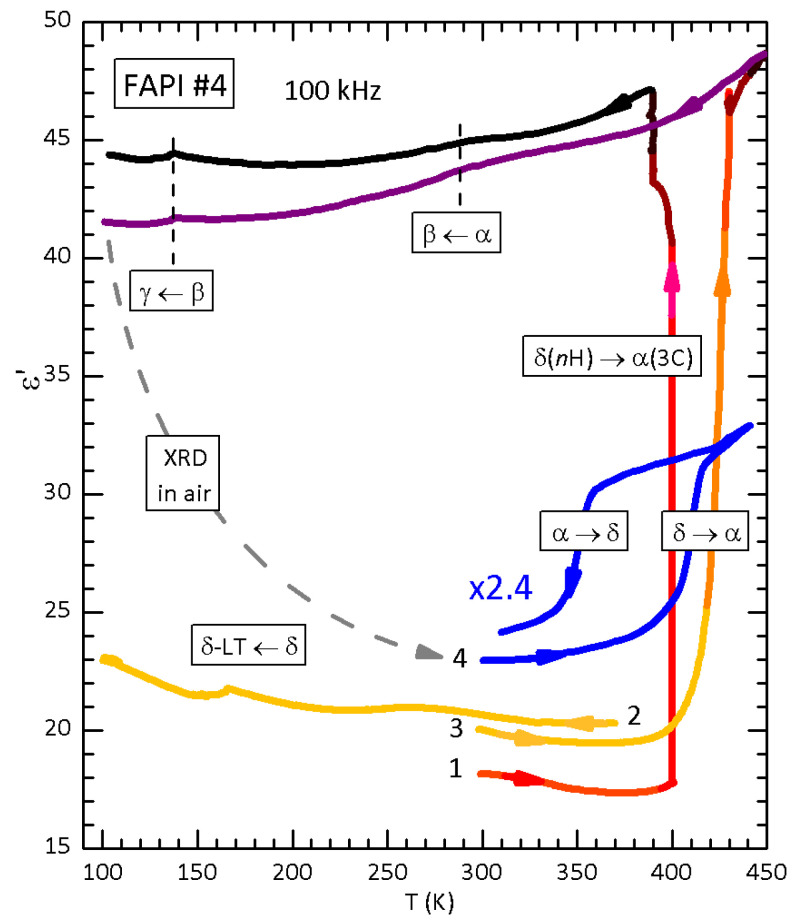
Dielectric permittivity of FAPI measured at 100 kHz. Curves 1–3 were presented in Figure 1 of Reference [19]. Curve 4 is rescaled for clarity, but it should start at $\varepsilon^{\prime }=18-20$. Adapted with permission from ref. [19]. Copyright (2020) American Chemical Society.

**Figure 5 nanomaterials-11-01610-f005:**
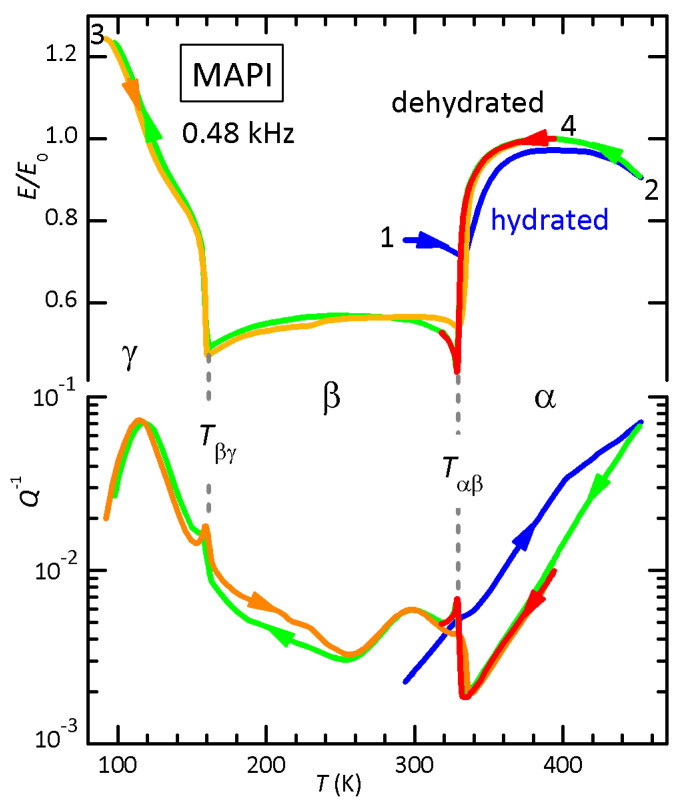
Relative change of the Young’s modulus and elastic energy loss of MAPI during temperature cycles in high vacuum. Curves 2–4 correspond to those in Figure 1 of [40]. Adapted with permission from ref. [40]. Copyright (2018) American Chemical Society.

**Figure 6 nanomaterials-11-01610-f006:**
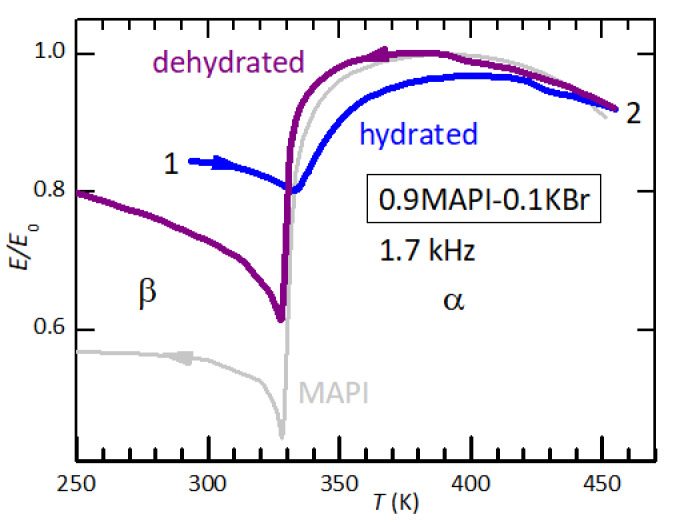
Young’s modulus of 0.9MAPI–0.1KBr measured in high vacuum in an initially hydrated condition. As for pure MAPI, the softening in the β phase is initially smaller (curve 1) than after dehydration (curve 2). With respect to pure MAPI (gray line), the amplitude of the step at the α/β transition in dehydrated condition is smaller, due to the fraction of inert KBr.

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
