# Peer review of "Structural Transitions and Stability of FAPbI3 and MAPbI3: The Role of Interstitial Water"

_nanomaterials, 2021, doi:10.3390/nano11061610_

Round 1
Reviewer 1 Report
The manuscript “Structural transitions and stability of FAPbI3 and MAPbI3: the role of interstitial water” by Francesco Cordero et al. discusses in depth the role of water on stability and phase transformation in MAPI and FAPI. The paper features aspects of a genuine article, an overview and discussion of own work, and elements of a review paper. The authors have the ambition to give a state of affairs on their understanding of said phase transitions, effects on stability and try to give a coherent picture to explain observations based on the use of mainly anelastic and dielectric spectroscopic measurements. It is not an easy article to read, takes quite an effort because of the many correlations and trends they want to discuss. The core claim is that interstitial water molecules are present and that they play a key role in catalyzing the transformations observed.
It is a relevant contribution to the community of hybrid perovskites and I favor publication in the journal “Nanomaterials” with minor revision.
Small remarks:
P1 introduction line 4: “3?”; I suppose the “?” has no meaning?
General comment: I would like the authors to comment on the reproducibility of this kind of measurements, I have the impression that every sample presented is quite unique, so reproducibility could be an issue.
General comment: The caption with a figure should be self-explanatory. That is not always the case, e.g. fig5
Reviewer 2 Report
The manuscript provides a study on the effects of water in the phase transitions of FA and MA lead halide perovskites, characterizing powder pellets by anelastic and dielectric spectroscopies. It is concluded that water molecules can diffuse in the perovskite lattice in an interstitial form and can catalyse the structural transformations between the different perovskite polymorphs. The study is interesting for the field, because the effects of water in the perovskite is still an open discussion, and the findings of the work are important to help explaining why water is beneficial during FAPbI3 film formation but bad for long-term stability.
However, the present work is based on “recycled” data from previous publications, the experiments of this work have not been designed specifically in the basis of the hypothesis. In addition, Sample FAPI#6, curve 3 in Figure 1, has been made of a mixture of new and old powders with unknown different phases and degrees of water adsorption, moreover the authors do not know whether it has been degraded to PbI2 or oxides. The analysis and interpretation of such a sample with several uncontrolled variables can lead to misleading conclusions, even if the main point is that it contains initially higher amounts of water than the other two samples. Moreover, a range of data around a critical point has been lost due to some incident, and is not clear whether the missing data would be relevant for the conclusions.
The work would be greatly improved and the conclusions would be stronger by measuring dedicated samples with similar composition and varying in a controlled way the exposure to humidity and also by doing the measurements at various heating rates and degassing conditions, including XRD to ensure that no decomposition occurred. Therefore I would recommend the manuscript for publication after measuring a new and controlled set of samples.
Moreover, the last part of the manuscript includes an extensive literature review to support their conclusions. The flow of the manuscript would be greatly enhanced and better digested by briefly listing main highlights of the sections 4.1 to 4.8.
Other comments:
Page 1 line 28: FAPbI3 is not the hybrid halide perovskite with smallest bandgap, as MASnI3 has 1.3eV. Specify that it has the smallest bandgap among the lead halide perovskites.
Page 1 line 33. Suggestion: updated citation demonstrating stabilization of a-FAPI with COOH- incorporation, with a record-efficiency and 450h of stable operation. https://doi.org/10.1038/s41586-021-03406-5
Line 75: In this paragraph the preparation of the pressed discs is described. Two sets of samples are described imprecisely as “old samples” and “unsuccessful pressing and re-homogenization”, how old? Stored in what conditions? What preparation conditions for repressing the disc? The authors state that new grain boundaries are formed, increasing the hygroscopicity, but no evidences are shown.
Figure 1 shows data from a previous publication of the authors, the necessary rights and permissions statement from ACS publications needs to be included in the text.
Figure 5: grey curve is not labelled.
Line 524 : were -> where
Round 2
Reviewer 2 Report
The authors have addressed my comments and I recommend to publish the manuscript in its current form.